# Fungi in Microbial Culture Collections and Their Metabolites

Alexander Vasilenko, Natalya Ivanushkina, Galina Kochkina and Svetlana Ozerskaya *

All-Russian Collection of Microorganisms (VKM), Pushchino Scientific Center for Biological Research of the Russian Academy of Sciences (PSCBR RAS), G.K. Skryabin Institute of Biochemistry and Physiology of Microorganisms Russian Academy of Sciences (IBPM RAS), 142290 Pushchino, Russia; vasilenko@ibpm.pushchino.ru (A.V.); nei@ibpm.pushchino.ru (N.I.); gak@ibpm.pushchino.ru (G.K.)
* Correspondence: smovkm@gmail.com; Tel.: +7-4997832402

**Abstract:** This study presents the results of a comparative analysis of the fungal diversity in the world system of microbial culture collections on one side with a variety of known fungal producers on the other side. The main VKM databases used are FungalDC and Metabolites of Fungi and the central point of analysis is the fungal ability to synthesize promising metabolites for applied use. It indicates that the option of obtaining new promising strains from the collection funds is still underestimated by the scientific community. In particular, it is shown that no more than 3% of the total fungal species fund contained in culture collections are used practically. It is possible that their use will considerably expand the range of studied strains and lead to the acquisition of new scientifically significant data.

**Keywords:** collections of microorganisms; databases; fungal diversity; metabolites



## 1. Introduction

Fungi belong to a kingdom of living organisms with extremely high diversity. According to expert estimates, the number of fungal species currently ranges from 2.2 to 3.8 million. In recent decades, the rate of description of new taxa has increased significantly due to the fast advances in molecular-biological diagnostics [1]. The Mycobank (www.mycobank.org), the premier reference platform for mycology, lists the names of over 440,000 legitimately described species (including synonyms), mostly held in culture collections worldwide, a large part of them in the leading CBS-KNAW Culture Collection (Westerdijk Fungal Biodiversity Institute; https://wi.knaw.nl/page/Collection). Obviously, ex situ conservation of microbial diversity for use in fundamental and applied scientific developments is of great importance. At the same time, culture collections play a decisive role in providing researchers with reliable biological material, which is the basis of any high-quality scientific work [2].

The large Microbial Culture Collections (mCCs) and MicroBiological Resource Center (mBRCs) maintain significant holdings of biological material and related information [3] to facilitate access to the biological resources conserved. They ensure the availability of microorganisms for the further use in sustainable scientific development. The mBRCs presents software for searching the data of required strains in their databases using various parameters. This helps to visualize and analyze the available information, and to make it accessible for the users of the online system.

The diversity of mCC and mBRC mycobiota in collections gets great attention, since fungi and their metabolites may represent an alternative to many currently used chemical compounds in the future [4]. Various new natural substances with promising potential for biological, medical, and industrial applications can be isolated and identified from fungi. The importance of fungal secondary metabolites for biotechnology cannot be overestimated. They can have antimicrobial activity, be enzyme inhibitors, be growth hormones, etc. [5,6].

A wide taxonomic diversity of various collections makes it possible to find strains capable of biosynthesis of specific organic substances. However, experience shows that

the main fungal group studied by numerous researchers is very limited. Thus, in analysis of 245 patents related to the production of secondary metabolites and biotransformation processes using endophytic fungi [7], it was discovered that the most frequently mentioned fungi belong to the genera *Aspergillus*, *Fusarium*, *Trichoderma*, *Penicillium*, and *Phomopsis*. The representatives of these genera are also used in biomedicine, agriculture, and the food industry. Meanwhile, the biotechnological potential of other fungal groups can also be of high importance. They are, however, excluded from the scope of the research for a variety of reasons.

Therefore, for a global assessment of the biotechnological potential of fungal strains maintained in collections, it is necessary to collect all available and newly received information about their properties in specialized databases [8].

A recent analytical comparison of All-Russian Collection of Microorganisms (VKM) databases with the database ChEBI (Chemical Entities of Biological Interest; https://www.ebi.ac.uk/chebi/) and database ChEMBL (Chemical Database of European Molecular Biology Laboratory; https://www.ebi.ac.uk/chembl/) containing information on fungal metabolites showed that VKM have significant number of potentially interesting strains for a comprehensive study of their metabolome [9].

Due to the constant interest in the search for new producers of biotechnologically promising fungal metabolites, the specialized databases on the diversity of fungi maintained in collections worldwide (FungalDC—Fungal Diversity in Culture Collections) and the ability of fungi to produce secondary metabolites (Metabolites of Fungi) were constructed in VKM.

The goal of this study was to compare on the basis of VKM databases, the diversity of the total fund of fungi in collections worldwide with the diversity of known fungi-producers that synthesize metabolites promising for applied use.

## 2. Materials and Methods

Materials contained in databases FungalDC and Metabolites of Fungi were used in this research. Each database was constructed with the appropriate structure of tables, forms, and queries in Access 2010.

The FungalDC database [10–12] is available online at the VKM website (www.vkm.ru). A hyperlink to this database is provided on the Mycobank portal in the section "External links—Specimens and strains links" for each taxon mentioned on its pages. It should be noted that this database includes only those collections that have available electronic or printed catalogs of their holdings.

The table of the fungal diversity in the culture collections presented in the FungalDC has the following fields:

- Code.
- Country.
- Collection Acronym.
- WDCM Number.
- Full Name of Culture Collection.
- Count of Species.
- Genus.
- Species.
- Variant/Subspecies.

The Metabolites of Fungi database [13] is also available on the VKM website, but only in test mode so far.

The table of the fungal metabolites' diversity contains the following fields:

- Code.
- Name of database.
- Database ID.
- Name of Metabolite.
- Genus.

- Species.
- Strain Number.
- Reference.

The method used here is a comparison of the common data fields with the same values of Genus and Species in two databases. Additional information was obtained from the other fields: the distribution of taxa of the cumulative fungal fund in various collections of different countries; the diversity of unique chemical compounds—fungal metabolites; the number and diversity of the mentioned specific strains.

## 3. Results

The FungalDC database was analyzed to obtain information on the different types of fungi found in the world's culture collections (Table 1).

**Table 1.** Volume of information in FungalDC (on 2 June 2022).

| Characteristics | Volume of Information |
| --- | --- |
| Number of records | 84,276 |
| Number of countries | 53 |
| Number of culture collections | 279 |
| Number of fungal genera | 4799 |
| Number of fungal species | 32,495 |

Curation is an essential aspect of FungalDC that distinguishes it from WDCM (http://ccinfo.wdcm.org/). This indicates that virtually every taxon was validated against the Index Fungorum (http://www.indexfungorum.org/) and Mycobank databases. This makes it possible to achieve correspondence in the spelling of the mentioned taxa, and this simplifies the search of the desired taxon by users. In FungalDC, the numerous misspellings and errors of genera names and species epithets that could be found in catalogues of any level collection were corrected. Continued curation work will contribute to the compilation of the most correct list of species of the world fungal fund in FungalDC.

Most of the collections analyzed in FungalDC are WDCM-registered and belong to the World Federation of Culture Collections (WFCC). WFCC maintain information on collections of microorganisms from various countries and accumulate information about the species presented in them, and promote and support the establishment of new culture collections and related services (https://wfcc.info/about_view). The remaining collections included in the database were created at scientific institutions and do not have WDCM registration numbers; however, they publish catalogs of fungal cultures stored in them.

The FungalDC query system allows to see the abbreviations of culture collections, their names, the country where the collection is located, and the number of fungal species in each.

Table 2 presents the number of culture collections per country. The biggest numbers of them are in Asia (such as Thailand, Japan, and India). A complete list of collections that make their catalogues available to users is provided in Appendix A.

**Table 2.** Location of the collections by country (on 2 June 2022).

| Countries | Number of Culture Collections |
|---|---|
| Argentina | 3 |
| Armenia | 1 |
| Australia | 19 |
| Belarus | 1 |
| Belgium | 6 |
| Brazil | 32 |
| Bulgaria | 2 |
| Canada | 18 |
| Chile | 1 |
| China | 5 |
| Czech Republic | 11 |
| Denmark | 2 |
| Finland | 2 |
| France | 4 |
| Germany | 4 |
| Greece | 3 |
| Hong Kong | 1 |
| Hungary | 2 |
| India | 13 |
| Indonesia | 4 |
| Iran | 2 |
| Ireland | 1 |
| Italy | 3 |
| Japan | 15 |
| Malaysia | 2 |
| Mexico | 9 |
| The Netherlands | 1 |
| New Zealand | 5 |
| Pakistan | 1 |
| Philippines | 3 |
| Poland | 4 |
| Portugal | 1 |
| Republic of Korea | 1 |
| Romania | 1 |
| Russian Federation | 10 |
| Senegal | 1 |
| Serbia | 1 |
| Singapore | 2 |
| Slovak Republic | 3 |
| Slovenia | 2 |
| Spain | 2 |
| Sri Lanka | 1 |
| Sweden | 3 |
| Switzerland | 1 |
| Taiwan | 1 |
| Thailand | 36 |
| Turkey | 2 |
| UK | 9 |
| Ukraine | 2 |
| USA | 17 |
| Uzbekistan | 1 |
| Vietnam | 1 |
| Zimbabwe | 1 |

The information on the fungal species in the collection catalogues indicates that the total collection fund is more than 4700 genera and 32,000 unique species, including synonyms. Most fungal species are represented in European collections (Figure 1). This is

largely due to the capability of the world's leading mycological collection—CBS—in The Netherlands, which contains over 18,000 fungal species.

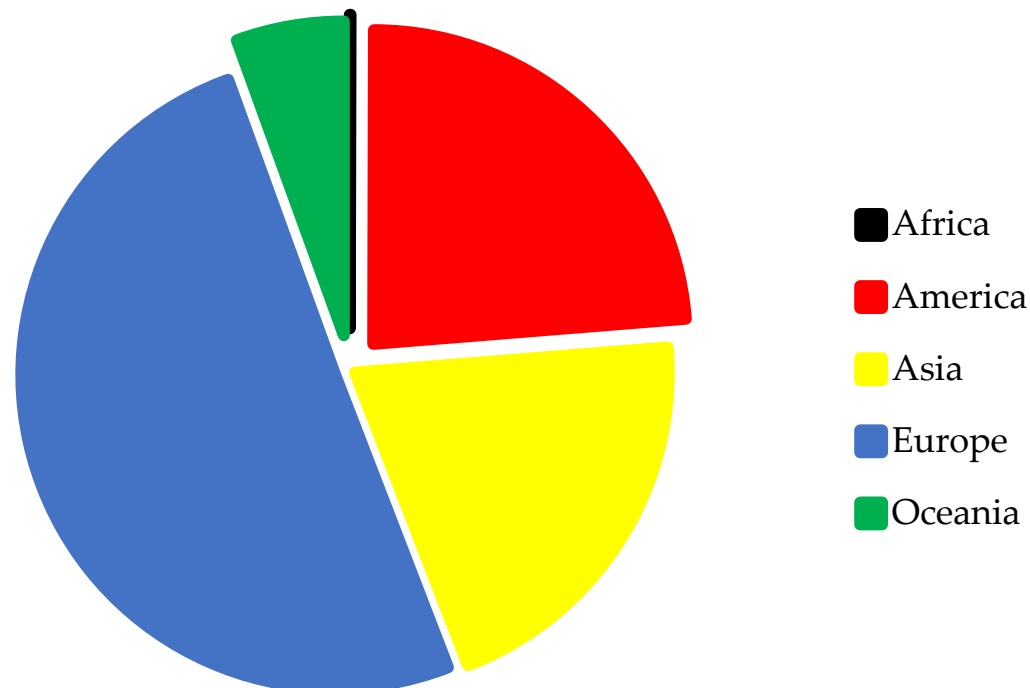

**Figure 1.** Distribution of fungal species in culture collections on different continents.

The genera whose strains are the most frequently isolated from different habitats and are widely studied in connection with their beneficial or harmful properties are the most numerous in the culture collections: *Penicillium*, *Candida*, *Aspergillus*, *Fusarium*, *Alternaria*, *Chaetomium*, *Colletotrichum*, *Phoma*, *Diaporthe*, and *Cladosporium*.

It should be noted that representatives of some fungal genera are preserved in the majority of the collections, or at least in a significant part of them (Table 3). These are, first of all, fungi whose metabolome is actively studied (*Penicillium*, *Fusarium*), yeasts actively used in the food industry (*Saccharomyces*), and opportunistic yeasts such as *Candida* and *Cryptococcus*. It is these fungi that are most in demand by users for scientific and practical research.

**Table 3.** Fungal genera preserved in the maximum number of collections (on 2 June 2022).

| Fungal Genera | Number of Culture Collections |
| --- | --- |
| *Penicillium* | 159 |
| *Saccharomyces* | 151 |
| *Fusarium* | 150 |
| *Candida* | 141 |
| *Trichoderma* | 135 |
| *Rhizopus* | 131 |
| *Mucor* | 127 |
| *Cryptococcus* | 124 |
| *Cladosporium* | 119 |
| *Chaetomium* | 110 |
| *Rhodotorula* | 109 |
| *Paecilomyces* | 105 |

However, there are collections, one of the tasks of which is to support fungi of rare taxa and to preserve biological diversity in general. They keep species that are poorly studied or new to science. This group also includes fungi isolated from extreme habitats.

These may be micromycetes with a high adaptive potential capable of active metabolism under unfavorable environmental conditions [14]. Such fungi are stored mainly in large bioresource centers such as CBS, MUCL, DAOMC, VKM, and others (Appendix A). It is in these poorly studied organisms that the potential of the kingdom of Fungi is concentrated, which has yet to be revealed.

The Metabolites of Fungi database was constructed on ChEBI (Chemical Entities of Biological Interest, (https://www.ebi.ac.uk/chebi/) and FungalMet (http://www.fungalmet.org/it/) databases.

The acronym ChEBI literally means "chemicals of biological interest database". It provides all researchers with open access to information on low-molecular-weight chemical compounds produced by fungi and reflects the relationships between individual chemicals, their families, and classes [15]. All this database information is of public access (Creative Commons license, CC BY 4.0). All the data presented have links to their sources. The main data sources for the ChEBI database are the databases: IntEnz, ChEMBL, KEGG COMPOUND, PDBeChem. Among the Life Sciences databases interacting with ChEBI, we found the following: ArrayExpress, EAWAG-BBD, BioModels, BRENDA, ChEMBL, ChemIDplus, COMe, DDBJ, DrugBank, EMBL, ENA, Enzyme Portal, Expression Atlas, GenBank, GMD, IEDB, IntAct, IntEnz, IUBMB, KEGG, KEGG DRUG, KEGG GLYCAN, LIPID MAPS, LMPD, LMSD, nmrshiftdb, NURSA, PDBe, PIR, PubChem, Reactome, RESID, Rhea, SABIO-RK, wwPDB, and UniProtKB.

The manual and default keyword search for fungal organisms in ChEBI provides comprehensive information on diverse fungal taxa and their metabolites.

FungalMet stores information on secondary metabolites of fungi that were addressed and correlated with fungal sources in scientific publications. Metabolites can be found using the search with various parameters, such as the microorganism-producer and the name of the compound or the chemical formula. Currently, the database contains more than 3000 metabolite names. When the new information comes, the database is updated with the new substances of the fungal origin. Information on all of its fungal taxa and on all the metabolite names was extracted from FungalMet. Interestingly, in the process of analyzing the data obtained, it turned out that these two bases largely complement each other, with very little overlap (Table 4).

**Table 4.** Data from different sources in the Metabolites of Fungi database (on 26 April 2022).

| Characteristics | Total | ChEBI | FungalMet |
|---|---|---|---|
| Number of records | 7437 | 4022 | 3226 |
| Number of unique metabolite records | 6397 | 3672 | 2725 |

Table 4 shows that these two databases together contain the information on more than 7400 metabolites. At the same time, 6397 of them are unique, that is, they are found only in one of them.

These metabolites are produced by 304 fungal genera, of which approximately 10% generate 20 or more metabolites and 35% produce a single metabolite. Several fungal taxa, however, as shown in Table 5, produce hundreds of beneficial compounds.

The yeast *Saccharomyces cerevisiae*, which has played an important role in food and beverage fermentation for centuries and has been extensively studied (Table 6), maintains a special place on the fungal list.

**Table 5.** Genera of fungi with high production of metabolites (on 26 April 2022).

| Fungal Genera | Number of Metabolites Produced by Members of the Genus |
|---|---|
| *Saccharomyces* | 1886 |
| *Aspergillus* | 1107 |
| *Penicillium* | 754 |
| *Ganoderma* | 434 |
| *Fusarium* | 293 |
| *Chaetomium* | 235 |
| *Alternaria* | 164 |
| *Trichoderma* | 128 |
| *Phoma* | 97 |
| *Acremonium* | 91 |

**Table 6.** Species of fungi with high production of metabolites (on 26 April 2022).

| Fungal Species | Number of Metabolites Produced by Members of the Species |
|---|---|
| *Saccharomyces cerevisiae* | 1881 |
| *Ganoderma lucidum* | 306 |
| *Aspergillus fumigatus* | 143 |
| *Aspergillus niger* | 128 |
| *Chaetomium globosum* | 108 |
| *Aspergillus terreus* | 85 |
| *Aspergillus flavus* | 53 |
| *Gibberella fujikuroi* | 53 |
| *Cordyceps sinensis* | 52 |
| *Aspergillus ochraceus* | 48 |
| *Claviceps purpurea* | 47 |
| *Penicillium citrinum* | 46 |
| *Aspergillus nidulans* | 42 |

As not all strains of the same species are equally active in the production of a particular metabolite, the substrate from which the strain was isolated, the duration and methods of its conservation in the collection, etc., are of high importance in research. In this regard, many scientific journals in their rules for authors indicate as compulsory the information on the number (designation) of the strain used in the study. For example, Journal of Microbiology in Instructions to Authors "strongly encourages authors to deposit important strains in publicly accessible culture collections and to refer to these collections and strain numbers in the manuscript".

Nevertheless, not all the Metabolites of Fungi database records with metabolites produced by micromycetes are accompanied by specific strain numbers. Additionally, only a very small part of them keep the records with the strain numbers of the known culture collections. As a result, only 1176 database records present the strains, the rest do not indicate it at all. Of these, there are 325 records with numbers of known collections (Table 7), they make 129 unique collection strains only. Some collections are excluded from the database because neither their catalogues nor their WDCM numbers are available.

The results of the comparative analysis on the diversity of the world collection fungal strains on one side with the diversity of fungi with the metabolites studied and presented in the most famous Life Sciences databases on the other side, are presented in Table 8.

**Table 7.** Strains in Metabolites of Fungi database (on 26 April 2022).

| Acronym of Culture Collection | Country | Website | Number in WDCM/Name of Collection | Records with the Strain Number | Number of Unique Strains |
|---|---|---|---|---|---|
| ATCC | USA | http://www.atcc.org/ | WDCM1 | 115 | 34 |
| BCC | Thailand | http://www.biotec.or.th/bcc/ | WDCM783 | 54 | 16 |
| CBMAI | Brazil | https://cbmai.cpqba.unicamp.br/?lang=en | WDCM823 | 1 | 1 |
| CBS | The Netherlands | http://www.westerdijkinstitute.nl/ | WDCM133 | 14 | 7 |
| CCTCC | China | http://www.cctcc.org/ | WDCM611 | 3 | 1 |
| DSMZ | Germany | http://www.dsmz.de/ | WDCM274 | 11 | 3 |
| IBT | Denmark | http://www.bioengineering.dtu.dk/english | WDCM758 | 4 | 2 |
| IFM | Japan | http://www.pf.chiba-u.ac.jp/ | WDCM60 | 8 | 2 |
| IFO (NRBC) | Japan | https://www.nite.go.jp/nbrc/catalogue/?lang=en | WDCM191 | 11 | 8 |
| IMI | UK | http://www.cabi.org/ | WDCM214 | 26 | 13 |
| KMM | Russia | http://www.piboc.dvo.ru/ | WDCM644 | 10 | 2 |
| NRRL | USA | https://nrrl.ncaur.usda.gov/ | WDCM97 | 54 | 30 |
| AJ | Japan | https://www.ajinomoto.com/ | Central Research Laboratories, Ajinomoto Co. Inc., Kawasaki, Japan | 5 | 3 |
| FERM | Japan | https://www.aist.go.jp/index_en.html | Patent and Bio-Resource Center, National Institute of Advanced Industrial Science and Technology (AIST), Tsukuba, Ibaraki, Japan | 3 | 3 |
| MRC | South Africa | www.samrc.ac.za | National Research Institute for Nutritional Diseases, Tygerberg, South Africa | 2 | 1 |
| NRCC | Canada | https://nrc.canada.ca/en | Division of Biological Sciences, National Research Council of Canada, Ottawa, Canada | 1 | 1 |
| TUF | Estonia | https://www.natmuseum.ut.ee/en/content/mycological-collection | The mycological collection of the University of Tartu, Estonia | 3 | 2 |

**Table 8.** Diversity of fungi in culture collections and metabolite database (on 2 June 2022).

| Name of Database | Number of Genera | Number of Species |
|---|---|---|
| FungalDC | 4799 | 32,495 |
| Metabolites of Fungi | 304 | 899 |
| Share (%) | 6.3 | 2.8 |

## 4. Discussion

The analysis of the integrated data made it possible to assess the extent to which fungi from different taxa have been studied in relation to their ability to produce metabolites and to understand in which collections of the world one or another known producer of certain chemical compounds should be sought.

The main interest in fungal metabolites is associated with the discovery of new drugs, since among the substances produced by fungi, most exhibit antibacterial, antifungal, or antitumor activity [16]. According to the literature data, these biologically active substances are found in certain taxa of fungi, mainly in the representatives of the *Pezizomycotina* subphylum and in several classes of basidiomycetes (for example, *Agaricomycetes* and *Exobasidiomycetes*) [17]. Our analysis of the database showed that, among the fungal genera whose representatives produce the maximum number of metabolites (more than 20), the greatest number actually belongs to the four classes of *Pezizomycotina* and *Agaricomycetes* of *Agaricomycotina* (Figure 2).

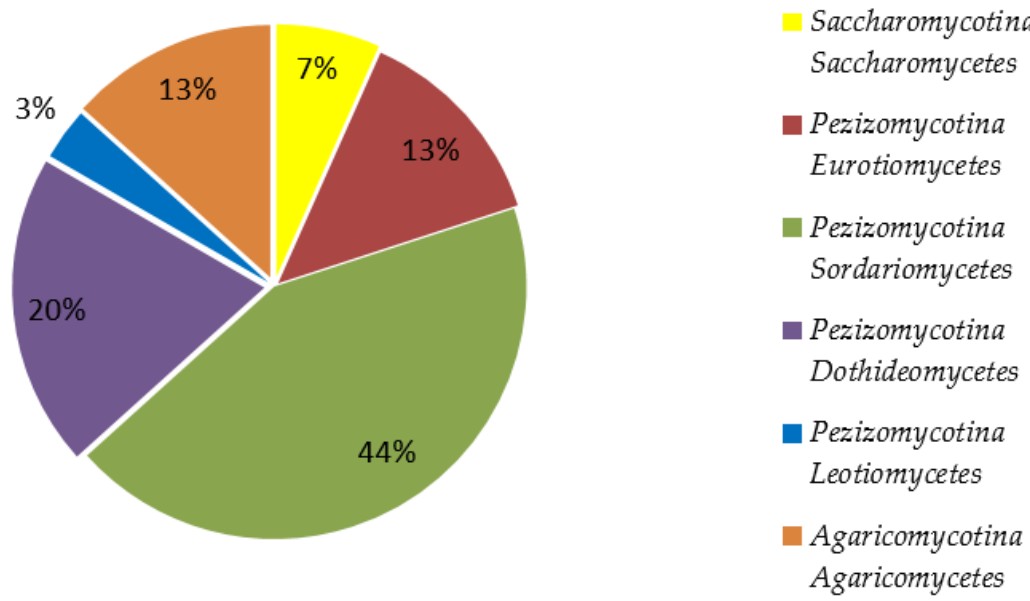

**Figure 2.** A variety of fungi with the greatest number of produced substances (more than 20).

The most inspected fungi belong to the order *Eurotiales* (*Eurotiomycetes*), including the genera *Aspergillus*, *Penicillium*, *Paecilomyces*, and *Paecilom*. Several dozen taxa are utilized in biotechnological research but are not supported in collections for *yces*, as well as certain fungal genera from the order *Hypocreales* (*Sordariomycetes*).

The last group has representatives of the genera *Fusarium*, *Trichoderma*, *Acremonium*, and others that have been constantly researched and studied over the years. Among the producers, there are also representatives of other phyla—*Mucoromycotina*, *Taphrinomycotina*, *Pucciniomycotina*, etc., but their number is disproportionately small. This is largely due to the insufficient use of the collection fund available for researchers.

Taxonomic diversity comparison in FungalDC and Fungal Metabolite database revealed only 70 fungal species known to be producers but not present in the culture collections.

Most of them are fungi whose metabolites were studied directly in the investigation of natural objects—a total of 50 species. These are difficultly cultivated basidiomycetes of the class *Pucciniomycetes* (*Glomospora* and *Uromyces*), pathogens of rust on cereal plants, lichenized ascomycetous fungi *Ramalina capitata* and *Pertusaria* sp., as well as representatives of the genera *Cytonaema* and *Smardaea* that form stromas on woody plants. This also includes the fungus *Plasmodiophora brassicae*, the causative agent of diseases of cruciferous plants, currently a representative of the Protozoa kingdom (*Plasmodiophoromycota*, *Plasmodiophoromycetes*). Most of the taxa of this group belong to the class *Agaricomycetes*

(23 genera, 41 species), a characteristic feature of which is the presence of rather large fruiting bodies, in the study of which metabolites were detected. Examples: the genera *Agaricus*, *Amanita*, *Boletus*, *Chlorophyllum*, *Clitocybe*, *Conocybe*, *Favolaschia*, *Ganoderma*, *Inocybe*, *Polyporus*, *Psilocybe*, *Tylopilus*, and others. Among ascomycetes, truffles *Tuber liyuanum* and *Tuber magnatum*, morel *Morchella importuna* can be included into this group.

The remaining 20 species are represented by cultivated micromycetes that are not maintained in collections according to the FungalDC database. Among them, the species with type strains not currently available and the species descriptions in the literature not sufficient to confirm the uniqueness of the taxon (for example, *Alternaria oryzae* [18], *Microascus tardifaciens* [19]), as well as taxa not represented in collections with available catalogs, such as *Pestalotiopsis fici*, *Pestalotiopsis oenotherae*, *Phomopsis paspali*, *Guanomyces polythrix*, *Cercospora coffeicola*, and *Sordaria araneosa*.

Several dozen taxa are utilized in biotechnological research but are not supported in collections for multiple reasons, as confirmed by the data obtained. On the other hand, the number of fungal taxa with the strains maintained in collections and with the known metabolites production is less than 3% of the total diversity of the total repository list of collections (Table 8).

On the one side, the diversity of the entire world fungal collection fund was compared to the diversity of the list of species traditionally utilized in scientific research. On the other side, it was demonstrated that the scientific community continues to underestimate the possibility of obtaining new promising strains from collection repositories.

The database FungalDC developed in VKM is available to users on various portals—www.vkm.ru and www.mycobank.org in online mode could possibly help with this issue. Perhaps its use will considerably expand the range of strains studied and lead to new scientifically significant data.

**Author Contributions:** Conceptualization, S.O. and A.V.; methodology, S.O. and A.V.; software, A.V.; validation, G.K. and N.I.; formal analysis, S.O.; investigation, A.V.; resources, S.O.; data curation, S.O., G.K. and N.I.; writing—original draft preparation, S.O., G.K. and N.I.; writing—review and editing, G.K. and N.I.; visualization, S.O.; supervision, S.O.; project administration, S.O. All authors have read and agreed to the published version of the manuscript.

**Funding:** This research has received funding from the Ministry of Science and Higher Education of the Russian Federation under grant agreement No. 075-15-2021-1051.

**Institutional Review Board Statement:** Not applicable.

**Informed Consent Statement:** Not applicable.

**Data Availability Statement:** Not applicable.

**Conflicts of Interest:** The authors declare no conflict of interest. The funders had no role in the design of the study; in the collection, analyses, or interpretation of data; in the writing of the manuscript, or in the decision to publish the results.

## Appendix A. List of Culture Collections (on 2 June 2022)

| | Country | Acronym | WDCM Number | Culture Collection Name | Number of Species |
|---|---|---|---|---|---|
| 1 | | BGIV | WDCM962 | Banco de Glomeromycota In Vitro (Bank of Glomeromycota In Vitro) | 8 |
| 2 | Argentina | CCM | WDCM29 | Coleccion de Cultivos Microbianos | 28 |
| 3 | | CEP | WDCM973 | Entomopathogenic Fungal Culture Collection of Argentina | 30 |
| 4 | Armenia | MDC | WDCM803 | Microbial Depository Center (National Microbial Culture Collection of the Republic of Armenia) | 305 |

|  | Country | Acronym | WDCM Number | Culture Collection Name | Number of Species |
|---|---|---|---|---|---|
| 5 |  | AMMRL | WDCM42 | Australian National Reference Laboratory in Medical Mycology | 326 |
| 6 |  | AWRI MCC | WDCM22 | AWRI Microorganism Culture Collection | 55 |
| 7 |  | CC | WDCM61 | CSIRO Canberra Rhizobium Collection | 1 |
| 8 |  | CS | WDCM532 | CSIRO Collection of Living Micro-algae | 1 |
| 9 |  | DE-CSIRO | WDCM70 | CSIRO Insect Pathogen Culture Collection | 14 |
| 10 |  | DMPMC | WDCM454 | Department of Microbiology | 20 |
| 11 |  | DFP | WDCM102 | DFP Culture Collection | 447 |
| 12 |  | FRR | WDCM18 | Food Science Australia, Ryde | 451 |
| 13 | Australia | WAITE | WDCM35 | Insect Pathology Pathogen Collection | 19 |
| 14 |  | JCT | WDCM387 | James Cook Townsville | 161 |
| 15 |  | KEMH | WDCM11 | KEMH/PMH Culture collection | 33 |
| 16 |  | ACH | WDCM47 | Mycology Culture Collection | 123 |
| 17 |  | WAC | WDCM77 | Plant Pathology Culture Collection | 347 |
| 18 |  | SBSFU | WDCM78 | School of Biological Sciences | 3 |
| 19 |  | SMTWA | WDCM90 | School of Medical Technology Western Australia | 1 |
| 20 |  | SAITP | WDCM569 | School of Pharmacy and Medical Sciences, University of South Australia | 1 |
| 21 |  | WACC | WDCM452 | Western Australian Culture Collection | 13 |
| 22 |  | WM | WDCM1205 | Westmead Medical Mycology Collection | 366 |
| 23 |  | DWT | WDCM36 | Wood Technology and Forest Research Division | 80 |
| 24 | Belarus | BIM | WDCM909 | Belarusian Collection of non-pathogenic microorganisms | 164 |
| 25 |  | MUCL | WDCM308 | Agro-food and Environmental Fungal Collection | 4601 |
| 26 |  | BCCM/IHEM | WDCM642 | BCCM/IHEM—Fungi Collection: Human and Animal Health | 1855 |
| 27 | Belgium | CRA-W | - | Fungi collection, Walloon Agricultural Research Centre | 11 |
| 28 |  | GINCO | - | Glomeromycota in vitro collection | 7 |
| 29 |  | LUC | - | Limburgs Universitair Centrum | 11 |
| 30 |  | CLO-Gent | - | *Verticillium chlamydosporium* (Fungi) strain collection | 1 |
| 31 |  | IPT | WDCM721 | Agrupamento de Biotecnologia, Culture Collection of Microorganisms | 4 |
| 32 |  | ITAL | WDCM143 | Banco de Fermentos Lacticos | 2 |
| 33 |  | CBMAI | WDCM823 | Brazilian Collection of Microorganisms from the Environment and Industry | 111 |
| 34 |  | BCCCp | WDCM921 | Brazilian Culture collection of Crinipellis perniciosa | 1 |
| 35 |  | CRM-UNESP | WDCM1043 | Central de Recursos Microbianos do Instituto de Biociencias da UNESP | 60 |
| 36 |  | FTI | WDCM716 | Centro de Biotecnologia e Quimica-CEBIQ | 14 |
| 37 | Brazil | CCB | WDCM713 | Colecao de Culturas de Basidiomicetos | 16 |
| 38 |  | CFAF | - | Colecao de Culturas de Fitopatogenos e Agentes de Controle Biologico de Fitopatogenos | 13 |
| 39 |  | Fiocruz/CCFF | WDCM720 | Colecao de Culturas de Fungos Filamentosos | 359 |
| 40 |  | CCT | WDCM885 | Colecao de Culturas Tropical | 692 |
| 41 |  | Fiocruz/CFAM | WDCM957 | Colecao de Fungos da Amazonia | 71 |
| 42 |  | CFEUnioeste | - | Colecao de Fungos Entomopatogenicos do Laboratorio de Biotecnologia Agricola | 5 |
| 43 |  | CFEOCA | - | Colecao de Fungos Entomopatogenicos Oldemar Cardim Abre | 19 |
| 44 |  | Fiocruz/CFP | WDCM951 | Colecao de Fungos Patogenicos | 20 |
| 45 |  | UFPEDA | WDCM114 | Colecao de Microrganismos UFPEDA | 127 |

| | Country | Acronym | WDCM Number | Culture Collection Name | Number of Species |
|---|---|---|---|---|---|
| 46 | | CICG | - | Colecao Internacional de Cultura de Glomeromycota | 24 |
| 47 | | CCMA-UFLA | WDCM1083 | Culture Collection of Agricultural Microbiology | 59 |
| 48 | | CCDCA | WDCM1081 | Culture Collection of Microorganisms from the Department of Food Science | 58 |
| 49 | | UFRJIM | WDCM725 | Departamento de Microbiologia Medica | 2 |
| 50 | | DPUA | WDCM715 | Departamento de Patologia/ICB | 81 |
| 51 | | IZ | WDCM724 | Departamento de Tecnologia Rural | 344 |
| 52 | | CG | WDCM712 | Embrapa Genetic Resources and Biotechnology Collection of Fungi of Interest to Biological Control | 42 |
| 53 | | CCOC | WDCM575 | Fundacao Oswaldo Cruz-FIOCRUZ | 45 |
| 54 | | CCT | WDCM711 | Fundacao Tropical de Pesquisas e Tecnologia "Andre Tosello" | 97 |
| 55 | | INPA | WDCM719 | Laboratorio de Micologia Medica Divisao de Microbiologia e Nutricao | 93 |
| 56 | | IALMIC | WDCM717 | Micoteca do Insituto Adolfo Lutz | 90 |
| 57 | | IMT | WDCM718 | Micoteca do Instituto de Medicina Tropical de Sao Paulo | 254 |
| 58 | | MGSS | - | Micoteca Prof. Gilson Soares da Silva | 61 |
| 59 | | CMRP | WDCM1240 | Microbiological Collections of Parana Network | 344 |
| 60 | | IAL | WDCM282 | Nucleo de Colecao de Micro-organismos | 3 |
| 61 | | ITALSM | WDCM723 | Secao de Microbiologia | 19 |
| 62 | | URM | WDCM604 | Universidade Federal de Pernambuco | 1144 |
| 63 | Bulgaria | BTCC | WDCM66 | Bulgarian Type Culture Collection | 77 |
| 64 | | NBIMCC | WDCM135 | National Bank for Industrial Microorganisms and Cell Cultures | 213 |
| 65 | | DAOMC | WDCM150 | Canadian Collection of Fungal Cultures | 2716 |
| 66 | | CSCC | - | Cereal Smuts Cultures Collection, Winnipeg Research Centre Agriculture and Agri-Food | 6 |
| 67 | | LSRRW | - | Department of Crop Sciences and Plant Ecology University of Saskatchewan | 2 |
| 68 | | MUL | WDCM250 | Department of Microbiology MUL-B 250 | 6 |
| 69 | | UWO | WDCM91 | Department of Plant Sciences | 294 |
| 70 | | HER | WDCM6 | Felix d'Herelle Reference Center for Bacterial Viruses | 1 |
| 71 | | DFF | WDCM50 | Forest Pathology Culture Collection, Pacific Forest Research Centre | 164 |
| 72 | Canada | FSC | WDCM237 | Fredericton Stock Culture Collection | 128 |
| 73 | | LYCC | WDCM634 | Lallemand Yeast Culture Collection | 1 |
| 74 | | OCRC | - | Oat crown rust Collection, Winnipeg Research Centre Agriculture and Agri-Food | 1 |
| 75 | | OSRC | - | Oat Stem Rust Collection, Winnipeg Research Centre Agriculture and Agri-Food | 1 |
| 76 | | PFCWDCC | - | PFC Wood Decay Culture Collection, Pacific Forestry Centre | 8 |
| 77 | | CCRCAF | - | Research Centre Culture Collection of Agriculture and Agri-Food | 2 |
| 78 | | SGSC | WDCM338 | Salmonella Genetic Stock Centre | 21 |
| 79 | | SCCM | WDCM920 | Sporometrics Culture Collection of Microorganisms | 30 |
| 80 | | UAMH | WDCM73 | UAMH Center for Global Microfungal Biodiversity | 1721 |
| 81 | | WLRC | - | Wheat Leaf Rust Collection, Winnipeg Research Centre Agriculture and Agri-Food | 1 |

| | Country | Acronym | WDCM Number | Culture Collection Name | Number of Species |
|---|---|---|---|---|---|
| 82 | | WSRC | - | Wheat Stem Rust Collection | 1 |
| 83 | Chile | CChRGM | WDCM1067 | Chilean Collection of Microbial Genetic Resources | 21 |
| 84 | | CCTCC | WDCM611 | China Center for Type Culture Collection | 822 |
| 85 | | ACCC | WDCM572 | Agricultural Culture Collection of China | 300 |
| 86 | China | CGMCC | WDCM550 | China General Microbiological Culture Collection Center | 1118 |
| 87 | | CCDM | WDCM117 | Culture Collection of Department of Microbiology | 41 |
| 88 | | CMCC(B) | WDCM123 | National Center for Medical Culture Collections | 34 |
| 89 | | RIBM | WDCM655 | Collection of Brewing Yeasts, Research Institute for Brewing and Malting | 28 |
| 90 | | CMF ISB | - | Collection of Microscopic Fungi ISB (CMF ISB) | 280 |
| 91 | | DBM | WDCM654 | Collection of Yeasts and Industrial Microorganisms, Institute of Chemical Technology, Department of Biochemistry and Microbiology | 103 |
| 92 | Czech Republic | DMUP | WDCM658 | Collection of Yeasts, Department of Genetics and Microbiology, Faculty of Science, Charles University | 71 |
| 93 | | CCBAS | WDCM558 | Culture Collection of Basidiomycetes | 288 |
| 94 | | CCDM | - | Culture Collection of Dairy Microorganisms | 16 |
| 95 | | CCF | WDCM182 | Culture Collection of Fungi | 615 |
| 96 | | RIFIS | - | Culture Collection of Microorganisms with Application in the Fodder Industry, Food Research Institute | 30 |
| 97 | | CCC | - | Czech Collection Clavicipitales | 31 |
| 98 | | CCM | WDCM65 | Czech Collection of Microorganisms | 513 |
| 99 | | CNCTC | WDCM130 | Czech National Collection of Type Cultures | 78 |
| 100 | Denmark | IBT | WDCM758 | IBT Culture Collection of Fungi | 103 |
| 101 | | SSI | WDCM158 | The International Escherichia and Klebsiella Centre (WHO) | 1 |
| 102 | Finland | HAMBI | WDCM779 | HAMBI Culture Collection | 36 |
| 103 | | VTTCC | WDCM139 | VTT Culture Collection | 139 |
| 104 | | UMIP | WDCM344 | Collection de Champignons et Actinomycetes Pathogenes | 229 |
| 105 | France | CNCM | WDCM174 | Collection Nationale de Cultures de Microorganismes | 54 |
| 106 | | LCP | WDCM659 | Fungal Strain Collection, Laboratory of Cryptogamy | 588 |
| 107 | | UCLAF | WDCM552 | HMR/Romainville | 34 |
| 108 | | BLWG | WDCM264 | Bayerische Landesanstalt fur Weinbau und Gartenbau | 41 |
| 109 | Germany | DSMZ | WDCM274 | DSMZ-Deutsche Sammlung von Mikroorganismen und Zellkulturen GmbH | 1603 |
| 110 | | IFAM | WDCM145 | Institut fur Allgemeine Mikrobiologie | 2 |
| 111 | | BBLF | WDCM204 | Institut fur Pflanzenschutz im Forst | 112 |
| 112 | | ATHUM | WDCM650 | ATHens University Mycology | 203 |
| 113 | Greece | BPIC | WDCM610 | Benaki Phytopathological Institute Collection | 127 |
| 114 | | NUA | WDCM281 | Department of Microbiology, National University of Athens | 1 |

|     | Country   | Acronym | WDCM Number | Culture Collection Name | Number of Species |
| --- | --------- | ------- | ----------- | ----------------------- | ----------------- |
| 115 | Hong Kong | CUHK    | WDCM68      | Biology Department, Chinese University of Hong Kong | 30 |
| 116 | Hungary   | DACT    | WDCM496     | Dept. Agricult. Chem. Technol. | 256 |
| 117 |           | NCAIM   | WDCM485     | National Collection of Agricultural and Industrial Microorganisms | 127 |
| 118 |           | MPKV    | WDCM448     | Biological Nitrogen Fixation Project College of Agriculture | 13 |
| 119 |           | CCDMBI  | WDCM119     | Culture Collection, Department of Microbiology | 78 |
| 120 |           | NTCCI   | WDCM107     | Culture Collection, Microbiology and Cell Biology Laboratory | 48 |
| 121 |           | DUM     | WDCM40      | Delhi University Mycological Herbarium | 1273 |
| 122 |           | DBV     | WDCM173     | Division of Standardisation | 1 |
| 123 |           | DMSRDE  | WDCM166     | DMSRDE Culture Collection | 150 |
| 124 | India     | UMFFTD  | WDCM562     | Food and Fermentation Technology Division, University of Mumbai | 5 |
| 125 |           | VPCI    | WDCM497     | Fungal Culture Collection | 60 |
| 126 |           | GPCK    | -           | Germplasm Centre for Keratinophilic Fungi | 22 |
| 127 |           | ITCC    | WDCM430     | Indian Type Culture Collection | 746 |
| 128 |           | MCM     | WDCM561     | MACS Collection of Microorganisms | 8 |
| 129 |           | MTCC    | WDCM773     | Microbial Type Culture Collection and Gene Bank | 1004 |
| 130 |           | NCIM    | WDCM3       | National Collection of Industrial Microorganisms | 288 |
| 131 |           | FNCC    | WDCM755     | Food and Nutrition Culture Collection | 92 |
| 132 |           | ICBB    | WDCM842     | ICBB Culture Collection for Microorganisms and Cell Culture | 27 |
| 133 | Indonesia | ITBCC   | WDCM44      | Institute of Technology Bandung Culture Collection | 57 |
| 134 |           | InaCC   | WDCM769     | Lembaga Ilmu Pengetahuan Indonesia, Indonesian Institute for Sciences | 150 |
| 135 | Iran      | IBRC    | WDCM950     | Iranian Biological Resource Center | 288 |
| 136 |           | PTCC    | WDCM124     | Persian Type Culture Collection | 50 |
| 137 | Ireland   | IMD     | WDCM227     | Industrial Microbiology Dublin | 104 |
| 138 |           | ITEM    | -           | Agro-Food Microbial Culture Collection | 92 |
| 139 | Italy     | CSMA    | WDCM147     | Centro di Studio dei Microorganismi Autotrofi—CNR | 1 |
| 140 |           | DBVPG   | WDCM180     | Industrial Yeasts Collection | 288 |
| 141 |           | AHU     | WDCM635     | AHU Culture Collection | 339 |
| 142 |           | OUT     | WDCM748     | Department of Biotechnology | 220 |
| 143 |           | ATU     | WDCM636     | Dept. of Biotechnology University of Tokyo | 4 |
| 144 |           | HUT     | WDCM195     | HUT Culture Collection | 241 |
| 145 |           | IAM     | WDCM190     | IAM Culture Collection | 509 |
| 146 |           | IFO     | WDCM191     | Institute for Fermentation, Osaka | 3024 |
| 147 |           | RIFY    | WDCM749     | Institute of Enology and Viticulture | 15 |
| 148 | Japan     | TIMM    | WDCM750     | Institute of Medical Mycology | 148 |
| 149 |           | JCM     | WDCM567     | Japan Collection of Microorganisms | 1984 |
| 150 |           | TSY     | WDCM67      | Laboratory of Mycology, Division of Microbiology | 5 |
| 151 |           | MAFF    | WDCM637     | NARO Genebank, Microorganism Section | 777 |
| 152 |           | NIBH    | WDCM746     | National Institute of Bioscience and Human-Technology | 15 |
| 153 |           | RIB     | WDCM640     | National Research Institute of Brewing | 14 |
| 154 |           | NRIC    | WDCM747     | Nodai Research Institute Culture Collection | 161 |

| | Country | Acronym | WDCM Number | Culture Collection Name | Number of Species |
|---|---|---|---|---|---|
| 155 | | IFM | WDCM60 | Research Center for Pathogenic Fungi and Microbial Toxicoses, Chiba University | 394 |
| 156 | Malaysia | SKUK | WDCM565 | Simpanan Kultur Universiti Kebangsaan | 40 |
| 157 | | UKKP | WDCM430 | Universiti Kebangsaan Kultur Perubatan | 21 |
| 158 | | CENACUMI | WDCM757 | Centro Nacional de Cultivos Microbianos (National Center For Microbial Cultures) | 277 |
| 159 | | CFQ | WDCM100 | Cepario de la Facultad de Quimica | 65 |
| 160 | | ITD | WDCM99 | Coleccion de Cepas Microbianas | 10 |
| 161 | | ENCB-IPN | WDCM449 | Coleccion de Cultivos de la Escuela Nacional de Ciencias Biologicas | 97 |
| 162 | Mexico | INIF | WDCM104 | Coleccion de Microhongos | 83 |
| 163 | | LIH-UNAM | WDCM817 | CultureCollection of Histoplasma capsulatum Strains from the Fungal Immunology Laboratory of the Department of Microbiology and Parasitology, Faculty of Medicine, UNAM | 1 |
| 164 | | IIBM-UNAM | WDCM48 | Industrial Culture Collection | 40 |
| 165 | | CDBB | WDCM500 | Unidad de Servicios de la Coleccion Nacional de Cepas Microbianas y Cultivos Celulares | 123 |
| 166 | | CISM | WDCM95 | *Verticillium dahliae* from cotton | 3 |
| 167 | The Nether-lands | CBS | WDCM133 | Centraalbureau voor Schimmelcultures, Fungal and Yeast Collection | 18,346 |
| 168 | | NZFS | WDCM62 | Forest Research Culture Collection | 100 |
| 169 | | ICMP | WDCM589 | International Collection of Microorganisms from Plants | 969 |
| 170 | New Zealand | WARC | WDCM376 | New Zealand Reference Culture Collection | 1 |
| 171 | | NZRD | WDCM318 | New Zealand Reference Culture Collection of Microorganisms, Dairy Section | 2 |
| 172 | | NZRM | WDCM457 | New Zealand Reference Culture Collection, Medical Section | 5 |
| 173 | Pakistan | FCBP | WDCM859 | First fungal culture bank of Pakistan | 119 |
| 174 | | ITDI | WDCM503 | Industrial Technology Development Institute | 49 |
| 175 | Philippines | UPCC | WDCM310 | Natural Sciences Research Institute Culture Collection | 214 |
| 176 | | PNCM-BIOTECH | WDCM620 | Philippine National Collection of Microorganisms | 144 |
| 177 | | LOCK | WDCM105 | Centre of Industrial Microorganisms Collection | 38 |
| 178 | | IAFB | WDCM212 | Collection of Industrial Microorganisms | 159 |
| 179 | Poland | IAW | - | Research and Development Centre for Biotechnology Culture Collection | 17 |
| 180 | | LCC | WDCM231 | University of Warmia and Mazury in Olsztyn | 34 |
| 181 | Portugal | MUM | WDCM816 | Micoteca da Universidade do Minho | 125 |
| 182 | Republic of Korea | EFCC | - | Entomopatogenic Fungal Culture Collection | 58 |
| 183 | Romania | ICCF | WDCM232 | Collection of Industrial Microorganisms | 21 |
| 184 | | VKM | WDCM342 | All-Russian Collection of Microorganisms | 2100 |
| 185 | Russian Federation | VIZR | WDCM760 | Collection for plant protection, All-Russian Institute of Plant Protection | 14 |
| 186 | | KMM | WDCM644 | Collection of Marine Microorganisms of the Pacific Institute of Biorganic Chemistry of the Far-Eastern Branch of the RAS | 48 |

| | Country | Acronym | WDCM Number | Culture Collection Name | Number of Species |
|---|---|---|---|---|---|
| 187 | | IPP | - | Collection of Monoxenic Cultures of Arbuscular Mycorrhizal Fungi of the Institute of Plant Physiology RAS | 4 |
| 188 | | VNIISC | - | Culture Collection of the Institute of Agricultural Microbiology, Russian Academy of Agricultural Sciences | 143 |
| 189 | | IBC | - | Institute of Cell Biology RAS | 13 |
| 190 | | LE(BIN) | WDCM1015 | Komarov Botanical Institute Basidiomycetes Culture Collection | 672 |
| 191 | | VKPM | WDCM588 | Russian National Collection of Industrial Microorganisms | 733 |
| 192 | | DSB MSU | - | The Department of Soil Sciences Moscow State University | 35 |
| 193 | | RIA | WDCM337 | The Russia Research Institute for Antibiotics Culture Collection | 43 |
| 194 | Senegal | MAO | WDCM53 | Mircen Afrique Ouest | 1 |
| 195 | Serbia | ISS | WDCM375 | Collection of Bacteria | 1 |
| 196 | Singapore | DBS | WDCM510 | Department of Biological Culture Collection | 98 |
| 197 | | NUSDM | WDCM568 | Department of Microbiology | 24 |
| 198 | Slovak Republic | CCWDF | - | Culture Collection of Wood-destroying Fungi | 52 |
| 199 | | CCY | WDCM333 | Culture Collection of Yeasts | 394 |
| 200 | | RIVE | WDCM28 | Research Institute for Viticulture and Enology | 96 |
| 201 | Slovenia | MZKI | WDCM599 | Microbial Culture Collection of National Institute of Chemistry | 174 |
| 202 | | ZIM | WDCM810 | ZIM Collection of Industrial Microorganisms | 112 |
| 203 | Spain | CECT | WDCM412 | Coleccion Espanola de Cultivos Tipo | 394 |
| 204 | | CCMCU | WDCM599 | Culture Collection of Microorganisms | 182 |
| 205 | Sri Lanka | DMBUK | WDCM564 | Department of Microbiology | 44 |
| 206 | Sweden | CCUG | WDCM32 | Culture Collection University of Goteborg | 92 |
| 207 | | FCUG | WDCM651 | Fungal Cultures University of Goteborg | 507 |
| 208 | | UPSC | WDCM603 | Uppsala University Culture Collection of Fungi | 800 |
| 209 | Switzerland | CCTM | WDCM475 | Centre de Collection de Type Microbien | 19 |
| 210 | Taiwan | BCRC | WDCM59 | Bioresource Collection and Research Center | 1545 |
| 211 | | BSMB | WDCM491 | Bacteriology and Soil Microbiology Branch | 20 |
| 212 | | BCC | WDCM783 | BIOTEC Culture Collection | 399 |
| 213 | | NRPSU | WDCM679 | Department of Agro-industry, Faculty of Natural Resources | 36 |
| 214 | | ABKMI | WDCM698 | Department of Applied Biology, Faculty of Science | 12 |
| 215 | | DBKKU1 | WDCM687 | Department of Biology, Faculty of Science | 27 |
| 216 | Thailand | SWU2 | WDCM697 | Department of Biology, Faculty of Science | 4 |
| 217 | | DBMU2 | WDCM667 | Department of Biotechnology, Faculty of Science | 62 |
| 218 | | FTCMU | WDCM690 | Department of Food Science and Technology, Faculty of Agriculture | 10 |
| 219 | | DMST | WDCM707 | Department of Medical Sciences Culture Collection | 204 |
| 220 | | MPSU | WDCM492 | Department of Microbiology | 2 |
| 221 | | DMKKU1 | WDCM680 | Department of Microbiology, Faculty of Medicine | 2 |
| 222 | | DMMU3 | WDCM668 | Department of Microbiology, Faculty of Medicine Siriraj Hospital | 66 |

| | Country | Acronym | WDCM Number | Culture Collection Name | Number of Species |
|---|---|---|---|---|---|
| 223 | | DMKU | WDCM669 | Department of Microbiology, Faculty of Science | 57 |
| 224 | | NU | WDCM696 | Department of Microbiology, Faculty of Science | 2 |
| 225 | | PPKU1 | WDCM670 | Department of Plant Pathology, Faculty of Agriculture | 5 |
| 226 | | PPKU3 | WDCM672 | Department of Plant Pathology, Faculty of Agriculture | 19 |
| 227 | | PPKU4 | WDCM673 | Department of Plant Pathology, Faculty of Agriculture | 18 |
| 228 | | PPKU5 | WDCM674 | Department of Plant Pathology, Faculty of Agriculture | 10 |
| 229 | | MLMJI | WDCM701 | Department of Plant Protection, Faculty of Agricultural Production | 7 |
| 230 | | CMKKU | WDCM684 | Diagnostic Microbiology Unit Division of Clinical Laboratory Srinagarind Hospital, Faculty of Medicine | 9 |
| 231 | | IFRPD | WDCM676 | Institute of Food Research and Product Development, Kasetsart University | 35 |
| 232 | | KUFC | WDCM677 | Kasetsart University Fungus Collection, Department of Plant Pathology, Faculty of Agriculture | 18 |
| 233 | | KKU | WDCM23 | MICKKU Culture Collection | 18 |
| 234 | | MLLD | WDCM702 | Microbiological Research Laboratory, Soil and Water Section, Department of Land Development | 3 |
| 235 | | CHULA | WDCM511 | Department of Microbiology, Faculty of Science | 34 |
| 236 | | DMCU | WDCM663 | Microbiology Department, Faculty of Science | 24 |
| 237 | | MLRU | WDCM695 | Microbiology Laboratory, Department of Biology, Faculty of Science | 7 |
| 238 | | MSDS | WDCM494 | Microbiology Section, Biological Science Division, Department of Science Services | 9 |
| 239 | | MSCMU | WDCM692 | Microbiology Section, Chiang Mai University (MSCMU) | 50 |
| 240 | | MSPP | WDCM704 | Mycology Section, Plant Pathology and Microbiology Division, Department of Agricultural Science | 4 |
| 241 | | NCSC | WDCM664 | National Center of Streptococcus Collection, Department of Microbiology, Faculty of Medical Science | 16 |
| 242 | | PCU | WDCM662 | Pharmaceutical Sciences Chulalongkorn University Culture Collection | 10 |
| 243 | | PPKM | WDCM699 | Plant Production Technology Department, Faculty of Agricultural Technology | 5 |
| 244 | | ERAEP | WDCM706 | Radiation Ecology Section, Biological Science Division, Office of Atomic Energy for Peace | 5 |
| 245 | | SSMJI | WDCM700 | Science Section, Department of General Education, Faculty of Agricultural Business | 9 |
| 246 | | TISTR | WDCM383 | TISTR Culture Collection, Bangkok MIRCEN | 235 |
| 247 | Turkey | KUKENS | WDCM101 | Centre for Research and Application of Culture Collections of Microorganisms | 83 |
| 248 | | RSKK | WDCM828 | Refik Saydam National Type Culture Collection-RSKK | 22 |
| 249 | | IMI | WDCM214 | CABI Bioscience Genetic Resource Collection | 3716 |
| 250 | UK | BEG | WDCM777 | La Banque European des Glomales | 30 |
| 251 | | NCPF | WDCM184 | National Collection of Pathogenic Fungi | 151 |
| 252 | | NCTC | WDCM154 | National Collection of Type Cultures | 1 |

|  | Country | Acronym | WDCM Number | Culture Collection Name | Number of Species |
|---|---|---|---|---|---|
| 253 |  | NCWRF | WDCM134 | National Collection of Wood Rotting Fungi | 296 |
| 254 |  | NCYC | WDCM169 | National Collection of Yeast Cultures | 449 |
| 255 |  | PHBL | WDCM508 | Philip Harris Biological Ltd. | 27 |
| 256 |  | DMCCUS | WDCM478 | School of Biological Sciences Culture Collection | 17 |
| 257 |  | CCMF | WDCM766 | University of Portsmouth | 207 |
| 258 | Ukraine | IBK | WDCM1152 | Culture Collection of Mushrooms | 195 |
| 259 |  | UCM | WDCM1203 | Ukrainian Collection of Microorganisms | 195 |
| 260 |  | NRRL | WDCM97 | Agricultural Research Service Culture Collection | 742 |
| 261 |  | ATCC | WDCM1 | American Type Culture Collection | 5585 |
| 262 |  | ARSEF | WDCM112 | ARS Collection of Entomopathogenic Fungi | 366 |
| 263 |  | LMS | WDCM530 | Carolina Biological Supply Company | 67 |
| 264 |  | FGSC | WDCM115 | Fungal Genetics Stock Center | 21 |
| 265 |  | INVAM | - | International Culture Collection of VA Mycorrhizal Fungi | 76 |
| 266 | USA | MCC | - | Mushroom Culture Collection Department of Plant Pathology, The Pennsylvania State University, USA | 198 |
| 267 |  | DM | - | National Center for Agricultural Utilization Research | 54 |
| 268 |  | NCMA | WDCM2 | Provasoli-Guillard National Center for Marine Algae and Microbiota | 4 |
| 269 |  | BMP | - | Pryor Lab Culture Collection, The University of Arizona, USA | 119 |
| 270 |  | RMF | - | Rocky Mountain Fungus, Wyoming | 374 |
| 271 |  | DSC | WDCM849 | The Dicty Stock Center | 16 |
| 272 |  | UA | - | The University of Alabama | 4 |
| 273 |  | UBC | - | University of California at Berkeley | 1 |
| 274 |  | UM | - | University of Maine, Orono, Maine, USA | 14 |
| 275 |  | WVDH | WDCM411 | West Virginia Hygienic Laboratory | 13 |
| 276 |  | WSF | - | Wisconsin Soil Fungus | 183 |
| 277 | Uzbekistan | NCAM | WDCM808 | National Collection of Agricultural Microorganisms | 146 |
| 278 | Vietnam | CNTP | - | The Industrial Microorganisms Culture Collection | 39 |
| 279 | Zimbabwe | BDUZ | WDCM17 | Biological Sciences | 17 |

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
