# Peer review of "Fungi in Microbial Culture Collections and Their Metabolites"

_diversity, doi:10.3390/d14070507_

Round 1
Reviewer 1 Report
Dear Author,
The manuscript has been reviewed. Well written and well presented.
Minor correction in your statistics: Iran has 6 more fungal collections. ABRIICC, IRAN, MCI. ACC, IBRC
All the best
Author Response
Dear sir/madam,
Thank you very much for your comments.
We have checked all the Iranian collections you mentioned and can report the following. 3 collections (ABRIICC, IRAN, ACC) do have WDCM numbers but do not publish catalogues of their strains, so its cannot be included in the sample analysed. The MCI collection does not support fungi. For the IBRC collection, we were able to find a catalogue on the Iranian Biological Resource Centre website, although there is no link to it on the WDCM website. We have modified the data given in the article to take into account the IBRC collection.
Reviewer 2 Report
Mainly this paper overviews and focused on the databases and fungal collections around the world.
As you know, the definition for micromycetes in kinds of literature and encyclopedia is: Endophytic fungi are micromycetes that infect the living plant tissue internally without causing any disease.
According to that, the yeasts like fungi, Candida, and Cryptococcus are not considered micromycetes.
In Line 153, please change it to “opportunistic yeasts”
Author Response
Dear sir/madam,
Thank you very much for the comments made.
We certainly agree with the comment made regarding "opportunistic yeasts" and have amended line 153 accordingly. Regarding the term micromycetes, we would like to point out that in the mycological literature (according to “Ainsworth and Bisby's Dictionary of the Fungi”), it is applied to microfungi (not only endophytes) as opposed to the term macromycetes, which usually refers to cap mushrooms. It is in this context that we have applied the term.
Reviewer 3 Report
To authors, there are numerous issues which need to be addressed and recorrected. As a consequence, I advise a "manor revision" for the manuscript.
Please review the following information so that authors can improve the quality of the manuscripts here.
- Corrections to the MS suggested based on the page, line number, and suggested changes or modification as listed as following items:
Page 1, Line 29: Please change or add suggested information (>>) as follows:
CBS-KNAW Culture Collection
>>>
CBS-KNAW Culture Collection (Westerdijk Fungal Biodiversity Institute; https://wi.knaw.nl/page/Collection)
Pages 1-2, Lines 43-48: Please provide at least one or two citation(s) to support the aforementioned paragraph. As a result, please add it or them in the references part as well.
Various new natural substances with promising potential for biological, medical and industrial applications can be isolated and identified from fungi. The importance of fungal secondary metabolites for biotechnology cannot be overestimated. They can have antimicrobial activity, be enzyme inhibitors, be growth hormones, etc. A wide taxonomic diversity of various collections makes it possible to find strains capable of biosynthesis of specific organic substances (xxxxxx et al xxx; xxxxxx et al xxxxxx).
Page 2, Lines 50-51: Please provide at least one reference to support the aforementioned paragraph. As a result, please add it in the references part as well.
……..in analysis of 245 patents related to the production of secondary 50 metabolites and biotransformation processes using endophytic fungi (xxxx et al.xxxx); it was discovered…..
Page 2, Lines 61-62: Please change or add suggested information (>>) as follows:
….the database ChEBI (Chemical Entities of Biological Interest) and database ChEMBL (Chemical Database of European Molecular Biology Laboratory)…..
>>>
….the database ChEBI (Chemical Entities of Biological Interest; https://www.ebi.ac.uk/chebi/), and database ChEMBL (Chemical Database of European Molecular Biology Laboratory; https://www.ebi.ac.uk/chembl/).........
Page 3, Line 137: Please change or add suggested information (>>) as follows:
…of them are in Asia….
>>>
…of them are in Asia (such as Thailand, Japan and India)….
Page 4, Line 144: Please give the date, month, and year that you obtained under access to this information.
Table 2. Location of the collections by country
>>>>
Table 2. Location of the collections by country (date?/month?/year?)
Page 5, Line 149: Please change or add suggested information (>>) as follows:
Diaporthe, Cladosporium
>>>
Diaporthe, and Cladosporium
Page 6, Line 167: Please give the date, month, and year that you obtained under access to this information.
Table 3. Fungal genera preserved in maximum number of collections >>> Table 3. Fungal genera preserved in maximum number of collections (date?/month?/year?)
Page 6, Line 182: Please change or add suggested information (>>) as follows:
…wwPDB, UniProtKB.
>>>
…wwPDB, and UniProtKB.
Page 7, Line 201: Please give the date, month, and year that you obtained under access to this information.
Page 7, Line 206: Please give the date, month, and year that you obtained under access to this information.
Page 8, Line 224: Please give the date, month, and year that you obtained under access to this information.
Page 8, Line 224: May I suggest authors to add two more column as second and third ones to refer “the country name” and “its website”?
For example, please add “United States of America” and “https://www.atcc.org” as 2nd and 3rd column, respectively for ATCC as the first column (Table 7).
For instance, please add “Thailand” and https://tbrcnetwork.org” as 2nd and 3rd column, respectively for BCC as the first column (Table 7).
For example, please add “Estonia” and “https://www.natmuseum.ut.ee” as 2nd and 3rd column, respectively for TUF as the first column (Table 7).
Page 10, Lines 275-276: Please change or add suggested information (>>) as follows:
Cercospora coffeicola, Sordaria araneosa.
>>>
Cercospora coffeicola, and Sordaria araneosa.
Page 9, Line 239, Line 248: these are not division, it should be “sub-phyla”
subdivision
>>>
subphylum
other divisions
>>>
other subphyla
Page 11, Line 309: Please give the date, month, and year that you obtained under access to this information.
- Optionally, the page, line number, and suggested changes to the English writing (or Paraphrasing) are alternatively recommended as follows:
Page 2, Lines 55-56: Please change or add suggested information (>>) as follows:
But, for various reasons, 55 they are left outside the range of research.
>>>
They are, however, excluded from the scope of the research for a variety of reasons.
Page 3, Lines 112-113: Please change or add suggested information (>>) as follows:
3.1. The analysis of the FungalDC database was carried out to obtain data on the 112 diversity of fungi contained in the culture collections of the world (Table 1).
>>>
3.1 The FungalDC database was analyzed to obtain information on the different types of fungi found in the world's culture collections (Table 1).
Page 3, Lines 116-119: Please change or add suggested information (>>) as follows:
An important feature of FungalDC that distinguishes it from WDCM (http://ccinfo.wdcm.org/) is that it is curated. This means that virtually every taxon was verified with known nomenclature fungal databases - Index Fungorum (http://www.indexfungorum.org/) and Mycobank.
>>>
Curation is an essential aspect of FungalDC that distinguishes it from WDCM (http://ccinfo.wdcm.org/). This indicates that virtually every taxon was validated against the Index Fungorum (http://www.indexfungorum.org/) and Mycobank databases.
Pages 3-4, Lines 141-143: Please change or add suggested information (>>) as follows:
Most of species are presented in the European collections (Figure 1). This is largely due to the fact that the world's leading mycological collection - CBS is located in the Netherlands, supporting over 18,000 fungal species
>>>
Most of fungal species are represented in European collections (Figure 1). This is largely due to the capability of the world's leading mycological collection - CBS - in the Netherlands, which contains over 18,000 fungal species.
Page 8, Lines 221-223: Please change or add suggested information (>>) as follows:
Not all of the collections are included in the database, because they do not provide catalogues of their collections, and some do not have a WDCM number.
>>>
Some collections are excluded from the database because neither their catalogues nor their WDCM numbers are available.
Page 7, Lines 203-205: Please change or add suggested information (>>) as follows:
The yeast Saccharomyces cerevisiae, which play an important role in fermentation of food and beverage for centuries and were studied extensively (Table 6), occupy a special place in the list.
>>>
The yeast Saccharomyces cerevisiae, which has played an important role in food and beverage fermentation for centuries and has been extensively studied (Table 6), maintains a special place on the fungal list.
Page 6, Lines 171-174: Please change or add suggested information (>>) as follows:
ChEBI database name literally means the database of chemicals of biological interest. It provides an open access for all the researchers to information on low-molecular-weight chemical compounds produced by fungi, and reflects the relationships between individual chemicals, their families and classes
>>>
The acronym ChEBI literally means "chemicals of biological interest database". It provides all researchers with open access to information on low-molecular-weight chemical compounds produced by fungi and reflects the relationships between individual chemicals, their families and classes.
Page 6, Lines 183-184: Please change or add suggested information (>>) as follows
In ChEBI the manual and the default keyword search for fungal organisms gives a significant amount of information on various fungal taxa and their metabolites.
>>>
The manual and default keyword search for fungal organisms in ChEBI provides comprehensive information on diverse fungal taxa and their metabolites.
Page 6, Lines 185-186: Please change or add suggested information (>>) as follows
FungalMet keeps information about the secondary metabolites of fungi presented in the scientific publications, characterized by and associated with the fungal sources
>>>
FungalMet stores information on secondary metabolites of fungi that were addressed and correlated with fungal sources in scientific publications.
Page 7, Lines 198-200: Please change or add suggested information (>>) as follows
These metabolites are the products by 304 fungal genera, approximately 10% of them produce 20 or more metabolites and 35% produce one metabolite each. But there are also the genera (Table 5) that produce hundreds of beneficial substances.
>>>
These metabolites are produced by 304 fungal genera, of which approximately 10% generate 20 or more metabolites and 35% produce a single metabolite. Several fungal taxa, however, as shown in Table 5, produce hundreds of beneficial compounds.
Page 9, Lines 244-246: Please change or add suggested information (>>) as follows
The most inspected fungi belong to the order Eurotiales of the class Eurotiomycetes, including the genera Aspergillus, Penicillium, Paecilomyces, and also to the order Hypocreales of the class Sordariomycetes.
>>>
The most inspected fungi belong to the order Eurotiales (Eurotiomycetes), including the genera Aspergillus, Penicillium, Paecilomyces, and Paecilomyces, as well as certain fungal genera from the order Hypocreales (Sordariomycetes).
Page 10, Lines 277-278: Please change or add suggested information (>>) as follows
The data obtained have confirmed that there are several dozen taxa used in biotechnological research, but (for various reasons) not supported in collections.
>>>
Several dozen taxa are utilized in biotechnological research but are not supported in collections for a multiple reason, as confirmed by the data obtained.
Page 10, Lines 282-285: Please change or add suggested information (>>) as follows
In conducted comparison of the diversity of the total world collection fund of fungi on one side with the diversity of the list of species traditionally used in scientific research on the other side, it has been shown that the scientific community has still underestimated the possibility of obtaining new promising strains from collection repositories.
>>>
On the one side, the diversity of the entire world fungal collection fund was compared to the diversity of the list of species traditionally utilized in scientific research. On the other site, it was demonstrated that the scientific community continues to underestimate the possibility of obtaining new promising strains from collection repositories.
